# C2FAR: Coarse-to-Fine Autoregressive Networks for Precise Probabilistic Forecasting

**Shane Bergsma**     **Timothy Zeyl**     **Javad Rahimipour Anaraki**     **Lei Guo**
Huawei Cloud, Alkaid Lab Canada
{shane.bergsma,timothy.zeyl,javad.anaraki,leiguo}@huawei.com

## Abstract

We present coarse-to-fine autoregressive networks (C2FAR), a method for modeling the probability distribution of univariate, numeric random variables. C2FAR generates a hierarchical, coarse-to-fine discretization of a variable autoregressively; progressively finer intervals of support are generated from a sequence of binned distributions, where each distribution is conditioned on previously-generated coarser intervals. Unlike prior (flat) binned distributions, C2FAR can represent values with exponentially higher precision, for only a linear increase in complexity. We use C2FAR for probabilistic forecasting via a recurrent neural network, thus modeling time series autoregressively in both space and time. C2FAR is the first method to simultaneously handle discrete and continuous series of arbitrary scale and distribution shape. This flexibility enables a variety of time series use cases, including anomaly detection, interpolation, and compression. C2FAR achieves improvements over the state-of-the-art on several benchmark forecasting datasets.

## 1 Introduction

Probabilistic forecasting is the task of estimating a joint distribution over future values of a time series, given a sequence of historical values. As an important problem with many applications and an abundance of real-world data, probabilistic forecasting has unsurprisingly come to be dominated by deep learning methods [7]. Such methods typically fit a global sequence model to a dataset of related time series. To forecast a given series, future values are iteratively generated from one-step-ahead univariate output distributions. In this paper, we propose coarse-to-fine autoregressive networks (C2FAR), a general method for modeling distributions of univariate numeric data, and we show how C2FAR enables improved output distributions for probabilistic forecasting.

Such improvements are needed because time series offer challenges not present in applications like text and image modeling. First, dynamic range or *scale* of real-world time series can vary widely within a single dataset [55]. The typical solution is to normalize the data based on input historical values, but many time series have heavy tails [19], meaning even after normalization, future values must be modeled via distributions with unbounded support. Second, even within one dataset, time series may be discrete, continuous, or mixed, and each of these may be distributed in arbitrary, multi-modal ways, which vary over time. All of this makes it difficult for practitioners to encode input values and to select appropriate parametric forms for output distributions.

C2FAR allows efficient modeling of any univariate numeric output (§3). C2FAR extends prior work using categorical output distributions over binned representations [49, 19]. While the precision of prior methods is limited by the number of bins in the discretization, C2FAR achieves an exponential increase in precision through only a linear increase in the number of bin parameters. Like digits in the decimal number system (and unlike, say, roman numerals or simple tally marks), C2FAR represents scalar quantities hierarchically, treating numbers as multidimensional objects, with each successive dimension or *level* representing the original value at a progressively finer resolution (Fig. 1).

36th Conference on Neural Information Processing Systems (NeurIPS 2022).

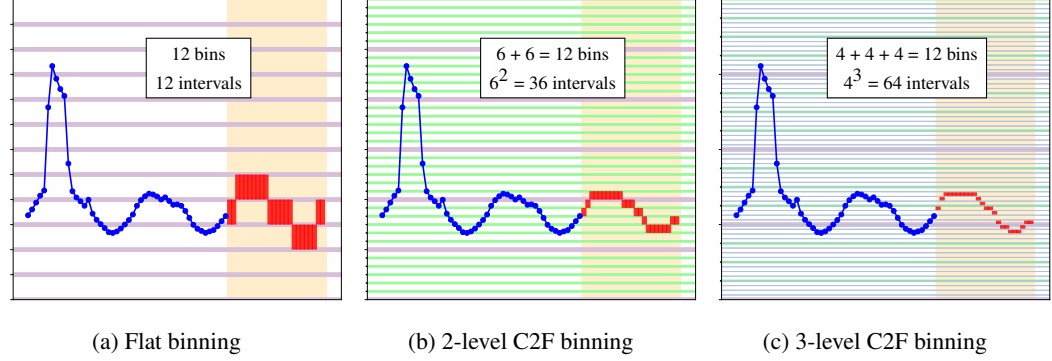

|                     |                        |                        |
|---------------------|------------------------|------------------------|
| (a) Flat binning    | (b) 2-level C2F binning| (c) 3-level C2F binning|

Figure 1: Flat vs. coarse-to-fine binnings for a *traff* time series (§5). C2F discretizations specify bin indices from coarse-to-fine, e.g. $3, 2, 2$. For the same total number of bins, the number of unique intervals in a C2F binning is much higher (and thus reconstruction error is much lower).

C2FAR proposes an autoregressive generative model for such representations. Levels are generated hierarchically from coarse to fine. Finer and finer intervals of support are generated from a sequence of categorical (binned) distributions, each distribution conditioned on the previously-generated coarser intervals. The conditional distributions are parameterized by a neural network, and the model is fit by maximizing log-likelihood of observed data. Compared to flat binnings, C2FAR models are more efficient for a given level of precision, and better enable learning of order and distance.

We use C2FAR to model output distributions in a deep forecasting model based on DeepAR [55] (§4). In our experiments, C2FAR-based forecasting models better recover synthetic distributions compared to recent state-of-the-art methods (§5.1). Evaluation on benchmark forecasting datasets shows that, in contrast to prior work, discretization improves accuracy (§5.2). Multi-level C2FAR models further improve over flat binnings across all datasets, including evaluation on a large new dataset of public cloud demand, which we release as a paper supplement. Code and data for C2FAR are available at https://github.com/huaweicloud/c2far_forecasting.

## 2 Background and related work

### 2.1 Probabilistic neural forecasting, use of binned output distributions

Probabilistic forecasting architectures include LSTMs [71, 55], temporal convolutions [58, 14] and transformers [37, 77, 72]. While C2FAR can be used with any such architecture, for simplicity we restrict our discussion in this paper to models that are autoregressive in time [37, 55]. While non-AR models typically output best-guess point forecasts [11, 58, 77] or specific quantiles of interest [71, 20], they can be made to output C2FAR parameters at all horizons (as was done with Gaussians in [14]).

Autoregressive networks must define step-wise input and output. To handle series of very different *scales*, input values are often normalized (e.g., dividing values by their mean [37, 55, 49]); output values are scaled back after prediction. A density over outputs is achieved by mapping network states to location and spread parameters of specific distributions. Practitioners must choose a distribution "to match the statistical properties of the data" [55]. In practice, Gaussian [55, 37], Student's-t [2], and Gaussian mixture distributions [46] have been used for real-valued data, while a negative binomial distribution [55] has been employed for discrete. Since output uncertainty is often not well captured by standard parametrics, recent work has investigated more flexible outputs, including spline quantile functions [24] and implicit quantile networks [28], which we compare to below (§5).

Rabanser et al. [49] use a categorical output distribution over a binned representation. Categoricals are flexible, but do not encode any underlying concept of bin order or distance (which must be learned). In prior work on pixel modeling, categorical distributions sometimes work better [69], sometimes worse [53] than mixture densities. In [49], the benefits of binning were mixed, and harmed DeepAR accuracy. Ehrlich et al. [19] "splice" a continuous Pareto distribution into a categorical in order to model extreme values; we show how Pareto distributions can also be used when samples are generated autoregressively using C2FAR (§3).

## 2.2 Challenges in forecasting mixed data, the density spike issue

Many time series are either fully discrete (e.g., counts), or continuous but with "clumps" at particular values. Dollar values are often specified to two decimal places. Even intrinsically continuous series are always processed or serialized to a certain precision; for example, one version of *elec* has been quantized to integers, another to six decimal places (§5). As a cloud provider, we observe many *semicontinuous* [42] series, with significant probability of being either 0% or 100%, while otherwise being continuous (*zero-inflation* is also common in fully discrete demand data [13, 57]).

Modeling discrete data can be challenging. The inherent limits of discrete series are rarely known at prediction time, so unbounded distributions such as negative binomials must be used [61]. Unfortunately, it is "not possible" for such distributions to output values in a normalized domain [55], precluding valuable scale-normalization. But since real-valued forecasts are usually acceptable for discrete data, practitioners often simply normalize discrete data and train continuous models. Gouttes et al. [28] report better results on *wiki* with a Student's-t distribution than with a negative binomial. Rabanser et al. [49] normalize *wiki* prior to "discretizing" it again for their categorical. We have also encountered practitioners with the (erroneous) belief that normalizing makes discrete data continuous.

Unfortunately, fitting powerful continuous models to discrete/mixed data "can lead to arbitrarily high density values, by locating a narrow high density spike on each of the possible discrete values" [68]. While this is a known issue in image modeling [66], it is apparently not well appreciated in forecasting. Density models based on splines [24] or flows [52] should therefore not be trained directly to maximize likelihood, as this may guide parameters (e.g., spline *knot* positions) solely to enable density spikes. A potential solution is *dequantizing*, e.g., adding `Uniform[0,1]` noise as in [52]. But this assumes: (1) we know the discrete series a priori, and (2) the resulting loss in precision does not outweigh the benefits of continuous models. For C2FAR, the ability to locate high-density spikes on discrete values is a feature, not a bug. C2FAR does not generate *arbitrarily* narrow spikes, rather it is restricted by the precision of the hierarchical binning. C2FAR can therefore train for likelihood on the actual discrete data, and it can generate realistic samples reflecting the true data distribution.

## 2.3 Coarse-to-fine representations, hierarchical softmaxes

Coarse-to-fine binnings have been used previously in machine learning. A coarse-to-fine representation is induced by algorithms that recursively discretize continuous-valued attributes for decision tree classifiers [21]. On the output side, SSR-Net [74] classifies human age in a coarse-to-fine (but not autoregressive) manner. $i$SAX [59] uses a coarse-to-fine representation for indexing time series.

Prior autoregressive models of discrete data can be viewed as instances of C2FAR. WaveRNN [33] models 16-bit audio samples by generating first the coarse (high) 8-bits, then the fine (low) 8-bits, conditioned on the sampled coarse. Vipperla et al. [70] found a different split between coarse and fine bits resulted in lower cross-entropy. These approaches can be regarded as two-level C2FAR models with classifier conditionals. The multi-stage likelihood from [57] can be viewed as a three-level C2FAR model, with binary classification at level 1 and 2, and Poisson regression to generate final counts. C2FAR unifies these approaches and generalizes them to unbounded, mixed/continuous data.

Beyond numeric data, hierarchical decompositions of categorical output distributions have been used previously in language modeling [45, 44]. As with C2FAR, the key benefit is exponentially fewer computations than when predicting words with a flat softmax. Whether further efficiencies such as variable-length representations (based on Huffman coding) [41] could prove effective when modeling continuous and ordinal data with C2FAR merits further investigation.

## 3 C2FAR networks

The process of discretization first partitions the support of a continuous variable into a finite number of distinct *bins*. A value is discretized by mapping it to the index of its corresponding bin. With C2FAR, partitioning is hierarchical: the support is first divided into a number of coarse bins, each of these coarse bins is further divided into a number of finer bins, and so on (recall Fig. 1 in §1). C2FAR thereby discretizes a data point, $z$, into a vector of $B$ indices for each of the $B$ coarse-to-fine levels, $\mathbf{z} = (z^1, \ldots, z^B)$. More formally, C2FAR assumes a discretization function $d : \mathbb{R} \to (\mathcal{B}_1, \ldots \mathcal{B}_B)$, where each $\mathcal{B}_i$ has one of $|\mathcal{B}_i|$ distinct values. Function $d^{\leftarrow}$ is a kind of inverse of $d$, mapping a set of bin indices to an *interval* in the original real-valued domain, $d^{\leftarrow} : (\mathcal{B}_1, \ldots \mathcal{B}_B) \to (\mathbb{R}, \mathbb{R})$.

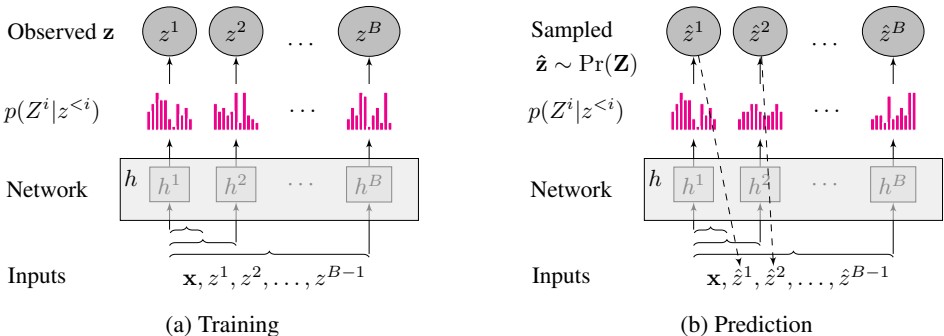

(a) Training    (b) Prediction

Figure 2: C2FAR network for regression. We train to minimize NLL of a set of examples; log loss on each example decomposes into the sum of NLL of each $z^i$ under its categorical distribution. In prediction, we sample from the categoricals, and autoregressively use the sampled values as inputs.

Let $\mathbf{z} = d(z)$ be the $B$-dimensional discretization of $z$. Let $\mathbf{z}^{<i}$ denote the first $i-1$ bin indices, $z^1, \ldots, z^{i-1}$. Let $\mathbf{x}$ be an optional set of features that may inform $z$'s distribution (e.g., for mixed data, $\mathbf{x}$ could flag which output bins contain integers). We follow prior work in modeling *multivariate* conditional distributions $p(\mathbf{z}|\mathbf{x})$ as autoregressive generative models via the chain rule [22, 6, 67]:

$$p(\mathbf{z}|\mathbf{x}) = \prod_{i=1}^{B} p(z^i|\mathbf{x}, \mathbf{z}^{<i}) \tag{1}$$

In C2FAR, each of the one-dimensional distributions, $p(z^i|\mathbf{x}, \mathbf{z}^{<i})$, is a categorical over the set of bin indices at the $i$th granularity, $\mathcal{B}_i$. Intuitively, the $i$th categorical is a distribution over the support *within* the generated $(i-1)$th bin. We parameterize these distributions using neural networks with a softmax output layer (specific architectures are discussed below), and fit the parameters to minimize negative log-likelihood (NLL) of training data (Fig. 2a). C2FAR is agnostic toward the encoding of the inputs $z^i$; they may, for example, be represented with 1-hot-encodings or embedding layers. As a generative model, we synthesize new data by first sampling a coarse bin ($\hat{z}^1$) given $\mathbf{x}$, then iteratively sampling finer and finer bins given the sampled (and encoded) coarser values (Fig. 2b).

C2FAR supports a variety of neural architectures: following prior work in multivariate distribution modeling, we may use a separate (feedforward) network for each categorical [6] (as in Fig. 2), or one global network with masking to enforce the autoregressive property [26]. Separate networks may share parameters across the levels [36] or with classifications at the same level across time (§4). Having separate networks for each level means the modeling problem at a given level need not change if we add more levels to the hierarchy; we adopt this approach in our forecasting experiments (§4.2).

While not strictly required, we use linear (evenly-spaced) binnings at each level. We hypothesize this enables the finer networks to better generalize learned concepts of order and distance (i.e., regardless of the coarse bins that they are conditioned on). We refer to the span from the first interval to the last interval as the *extent* of the binning. The extent, the number of levels, and the number of bins at each level, are C2FAR hyperparameters. We also consider the extreme high/low intervals to be open-ended, terminating at $\pm\infty$; we discuss the implications of this below.

**Level $B+1$ parametric distributions.** Let $(a,b)$ be the interval defined by $(a,b) = d^{\leftarrow}(\mathbf{z})$. To complete our generative story, we must generate a value within $(a,b)$ (rather than choosing a single "reconstruction value" for each bin as in [49]). Since our extreme intervals are open-ended, we cannot interpret our model as a piecewise uniform distribution [69], as uniforms are only defined on finite intervals. C2FAR solves this by allowing different distributions to be used depending on the interval $(a,b)$. Essentially, we assume a final conditional in (1) (implicitly at level $B+1$) that generates from a (differentiable) parametric distribution. We use distributions of the form:

$$p(Z^{B+1}|a,b) \sim \begin{cases} \text{Uniform}[a,b] & -\infty < a \text{ and } b < \infty \\ \text{Pareto}[a, \alpha_1] & b = \infty \\ -\text{Pareto}[b, \alpha_2] & a = -\infty \end{cases} \tag{2}$$

Here Pareto indicates a Type I Pareto distribution with fixed scale parameter ($a$ or $b$, defined a priori from the extent of the binning in the observed space) and dynamic shape parameter $\alpha_i$. Parameters

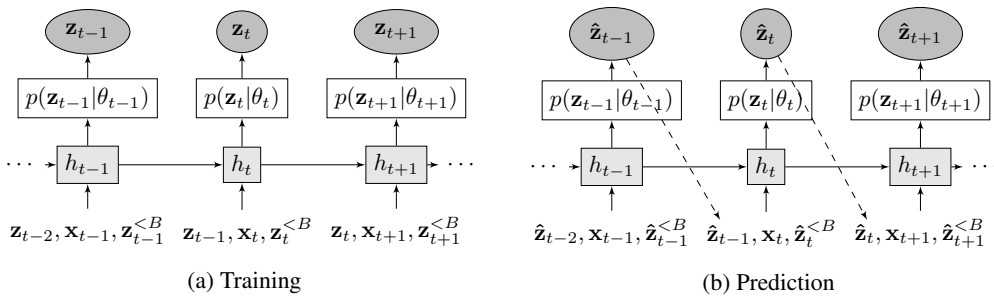

|                        |                        |
| :--------------------: | :--------------------: |
| (a) Training           | (b) Prediction         |

Figure 3: High-level C2FAR-RNN for forecasting. Unlike DeepAR, C2FAR-RNN uses discretized input/output vectors, $\mathbf{z}_t$, rather than scalars, and has additional (autoregressive) inputs $\mathbf{z}_t^{<B}$.

$\alpha_1$ and $\alpha_2$ are generated by the $h_{B+1}$ neural network in a manner analogous to how DeepAR outputs the mean and variance parameters of a Gaussian distribution [55].[1] We are effectively defining a piecewise-uniform density with Pareto-distributed tails.[2] We use this model for both continuous and discrete data. If data are truly discrete and precision is useful, our tuning procedure will choose more bins/levels. If one knows a priori the true discrete support, one may instead use discrete distributions at level $B + 1$, such as negative binomials, Poisson regression [57], or discrete uniforms.

**Complexity.** Consider a C2FAR discretization with $B$ levels and an unvarying cardinality of $K$ bins at each level; the original support of $z$ is effectively partitioned into $K^B$ total intervals, but modeled using only $KB$ categorical outputs in total ($B$ softmaxes with $K$ values each). We may regard C2FAR as a $K$-ary tree of height $B$. At each node in the tree, we determine which of the $K$ bins we fall into at that level, and follow the chosen branch to the sub-tree at the next level. We need only evaluate the softmax probabilities (and backpropagate gradients) for the $B$ nodes on the path from the root to the leaf of the tree. Since the height of the tree, $B$, is logarithmic (in base $K$) over the total number of intervals, we compute exponentially fewer outputs with C2FAR compared to flat binnings. The final likelihood additionally requires computing the probability at the $B + 1$ level according to the corresponding parametric distribution (Uniform or Pareto) as given in Eqn. (2). In this way, full C2FAR densities are never explicitly manifested when training or predicting; for plotting (e.g., Fig. 6), we explicitly compute likelihood at tiny increments over a predefined range.

## 4 Forecasting with C2FAR

We now explain how C2FAR can be used for $N$-step-ahead probabilistic forecasting. We base the forecasting framework on DeepAR [55], which has served as the basis for other recent improvements in forecast distribution modeling [24, 28], and therefore facilitates experimental comparison (§5).

### 4.1 DeepAR-style probabilistic forecasting

Let $z_t$ be the value of a time series at time $t$, and $\mathbf{x}_t$ be a vector of time-varying features or *covariates*. Probabilistic forecasting aims to model the conditional distribution of $N$ future values of $z_t$ (the *prediction range*) given the $T + N$ covariates, and $T$ historical values (the *conditioning range*):

$$p(z_{T+1} \ldots z_{T+N}|z_1 \ldots z_T, \mathbf{x}_1 \ldots \mathbf{x}_{T+N}) \stackrel{\text{def}}{=} p(z_{T+1:T+N}|z_{1:T}, \mathbf{x}_{1:T+N}) \tag{3}$$

---

[1]In this way, "weighted" Pareto tails are not "spliced" into the distribution at fixed user-defined quantiles, as they are in [19]. In C2FAR, Pareto tails provide the probability density function of values in the extreme bin, *given the values are in the extreme bin*. Each Pareto density itself integrates to 1, but is dynamically "weighted" by the probability of being in the corresponding extreme bin. Note also that if data does not have heavy tails, alternative distributions may be used in the extreme bins instead, e.g., left and right-truncated Gaussians.

[2]By parameterizing the quantile function with linear splines, SQF-RNN [24] actually enforces piecewise uniformity, over a *finite* range. Neural spline flows [18] require derivatives of the spline functions to match at knots, to avoid "numerical issues." By parameterizing the PDF directly using classifiers, and only manifesting the portion needed, C2FAR has none of the above restrictions, while being more efficient and well-behaved.

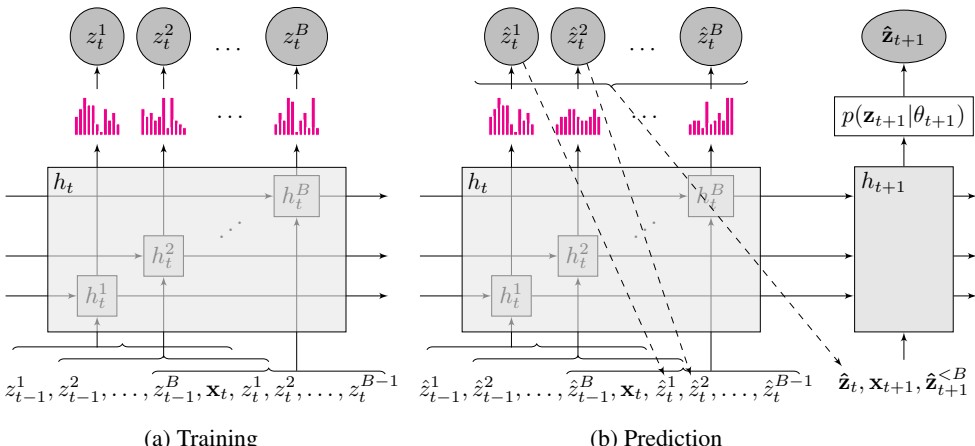

|  | (a) Training | | (b) Prediction |

Figure 4: C2FAR-RNN model detailing high-level unit, $h_t$, from Fig. 3. We train to minimize sum of NLL at each $z_t^i$ under a categorical with parameters given by a level-specific RNN, $h_t^i$. We predict by sequentially sampling a discretization at each time step, $\mathbf{z}_t$, which is re-input at step $t+1$.

DeepAR [55] formulates this conditional distribution as an autoregressive generative model:

$$p(z_{T+1:T+N}|z_{1:T}, \mathbf{x}_{1:T+N}) = \prod_{t=T+1}^{T+N} p(z_t|z_{1:t-1}, \mathbf{x}_{1:T+N}) = \prod_{t=T+1}^{T+N} p(z_t|\theta_t = y(h_t))$$

where $h_t = \mathrm{rnn}(h_{t-1}, z_{t-1}, x_t)$ is the output of an LSTM [30] recurrent neural network, and $y(\cdot)$ is a function that maps the output of the LSTM to the parameters of a parametric distribution $p(z_t|\theta_t)$. For example, $p(z_t|\theta_t)$ could be a Gaussian and $y(\cdot)$ output the Gaussian's mean and variance as $\theta_t$. Fig. 3a illustrates the overall architecture (with additions for C2FAR noted). Information about the conditioning range $z_1 \ldots z_T$ is conveyed through the state of the network at time $T$ (i.e., $h_T$). From an encoder-decoder perspective [63, 3], DeepAR uses the same network to encode and decode.

For training, each series is sliced into multiple *windows*, i.e., conditioning+prediction ranges at different start points. Windows are normalized using their conditioning ranges, and parameters are fit to minimize NLL of prediction-range outputs. To generate a forecast for a given (normalized) conditioning range, DeepAR draws samples $\hat{z}_{T+1} \ldots \hat{z}_{T+N}$ in sequence from $p(z_t|\theta_t)$ (Fig. 3b). Sampled roll-outs are unnormalized, and by repeating this procedure many times, a Monte Carlo estimate of (3) is obtained, from which desired forecast quantiles can be derived (see, e.g., Fig. 7).

## 4.2   Forecasting with C2FAR: C2FAR-RNN

We augment DeepAR by converting time series to their C2FAR discretization $\mathbf{z}_t = d(z_t)$. Rather than generating $\theta_t$ in one shot at each time step, we generate C2FAR categoricals autoregressively *within* each time step (Fig. 4). Each generated bin index, $z_t^{i-1}$, thus informs the distribution of the next bin index at that time step $z_t^i$. We also leverage information about discretized values at earlier time steps by replacing C2FAR's classifier-based conditionals (§3, Fig. 2) with LSTMs, one for each level in the C2FAR hierarchy. Intuitively, when generating bin index $z_t^i$, our inputs comprise both our current position in higher-level bins (e.g. $z_t^{i-1}$) *and* the $i$th-level index at the previous time step ($z_{t-1}^i$), along with the previous LSTM state, $h_{t-1}^i$.

We call the resulting system C2FAR-RNN. As with DeepAR, C2FAR-RNN is trained to minimize NLL of observed outputs in (normalized) prediction ranges (Fig. 4a). During training (and when evaluating NLL of test sequences), *at each time step*, all values are known and all distributions can be computed *in parallel*. When predicting (Fig. 4b), we must sample each bin index $z_t^i$ *sequentially*.

Recall that C2FAR has an implicit $B+1$ level, where a real value is generated from a parametric distribution (Eqn. (2)). We do not use an RNN for this level; the uniform distributions are fully-specified by the interval endpoints, while we generate Pareto parameters ($\alpha_1$, $\alpha_2$) via a simple feed-forward neural network (with a single hidden layer and softplus output transformation). To help inform the Pareto networks, we also encode real-valued $z_{t-1}$ as an additional input/covariate.

**Complexity**. Let $H$ be the number of RNN hidden units, $K$ a constant number of bins per level, and $B$ the number of levels ($I = K^B$ total intervals). The complexity-per-timestep of *each RNN* is essentially the sum of the RNN's recurrence operation, $H \times H$, and the projection of the recurrence to output bins, $K \times H$. For flat binnings, *overall* complexity is typically dominated by $K \times H$, while $B$-level C2FAR-RNN models are dominated by $B \times H \times H$, i.e., the cost of running $B$ RNNs in parallel. Measurements of timing and memory consumption (supplemental Table 8, 9) are well explained by these observations and corresponding values for H, K, and B (supplemental Table 6).[3]

**Tuning**. While C2FAR is trained for NLL, it is *tuned* for a given target metric. For forecasting, the metric is multi-step-ahead error. Thus we evaluate our forecasts during training by periodically running Monte Carlo sampling on our validation set and computing multi-step-ahead error. For tasks such as anomaly detection, denoising, etc., we would train for NLL and tune for an application-specific metric. While there is no one-size-fits-all metric for evaluating generative models, log-likelihood itself is sometimes regarded as the de facto standard [66]. Ironically, this is the one loss function C2FAR can *not* tune for, at least not on discrete data, because of the density spike issue (§2.2); if we tune directly for NLL, the tuner chooses more and more bins and levels, leading to narrower and higher spikes. If we *must* tune for log-likelihood, we could use discrete distributions in the $B + 1$ level. We could also add uniform noise, as other approaches do (but note in prior work this is required to enable *training*, not *tuning*, on discrete data). NLL evaluations are explored in supplemental §E.

### 4.3 Limitations and broader impact of C2FAR forecasting

Every B-level C2FAR model has an equally *expressive* [50] flat counterpart with a single categorical over all the fine-grained intervals; given unlimited training data, C2FAR may therefore not offer modeling benefits over flat binning. Conversely, when there is limited data, simple parametric distributions, with fewer parameters, may generalize better than C2FAR. Also, given it has extra hyperparameters (number of levels/bins-per-level), and given tuning search space grows exponentially with added hyperparameters, C2FAR may not discover optimum settings as quickly. We investigate experimentally whether the benefits of C2FAR outweigh these drawbacks on real-world data (§5.2).

C2FAR is more *complex* than flat binning, but whether it is less *efficient* depends on implementation. Compared to their 2-level coarse-to-fine model, WaveRNN [33] found flat binning required "significantly more parameters, memory and compute." We find C2FAR to run slower, but with less memory (Tables 8 and 9 in the supplemental). If flat binning was truly more efficient, we could always generate a large volume of data from a C2FAR model and *uptrain* [10, 48] a flat model on it.

**Broader impact.** By enabling simultaneous modeling of discrete and continuous series, without human involvement, C2FAR is a step toward a universal forecast model. Large C2FAR models could be trained on vast quantities of diverse series (similar to efforts in text [16, 39, 75] and vision [60, 64, 29]). Such models could be fine-tuned for new domains, and help make highly-accurate forecasting systems more widely-used, improving decision making and resource allocation.

There are also risks to this approach. Very large models have environmental and financial costs [4], which may be unnecessary when smaller models suffice. A universal model may be used more easily by those with less expertise and this may lead to misuse; for example, *automation bias* has been shown to disproportionately affect those with less domain expertise [9]. Recommended usage and possible misuse should be documented through artifacts such as model cards [43] and datasheets [25].

## 5 Experiments

C2FAR is implemented in `PyTorch` [47], using a 2-layer LSTM [30] with intra-layer dropout [62, 40], trained via Adam [34]. Notation C2FAR-RNN$_B$ refers to a $B$-level C2FAR model. We evaluate:

- C2FAR-RNN$_1$: essentially the standard flat-binning approach [49], but with Pareto tails [19]
- C2FAR-RNN$_2$: a two-level C2FAR-RNN model
- C2FAR-RNN$_3$: a three-level C2FAR-RNN model

---

[3]From a discretization perspective, the optimal $K$ is 2, as rather than a softmax, we may use a logistic function thresholded at 0.5 to select high vs. low bins, only computing $\log_2(I)$ total outputs. However, complexity also depends on the neural architecture. As C2FAR-RNN operates $B$ separate RNNs, it is more efficient to use a higher value of $K$ (but not so high that computing output probabilities dominates) and a lower value of $B$.

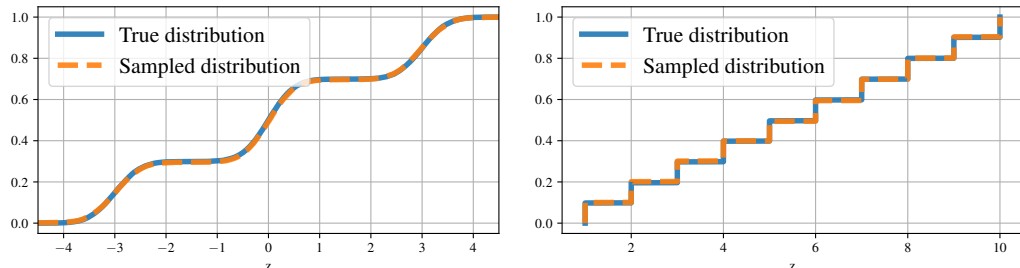

Figure 5: Distribution recovery: CDFs for a Gaussian mixture (left) and discrete uniform over $1 \ldots 10$ (right). After training C2FAR-RNNs on each, we sample paths and plot the *sampled* CDF. C2FAR recovers the true CDF much better than SQF-RNN or IQN-RNN (cf. Fig. 3 in [24], Fig. 2 in [28]).

## 5.1 Distribution recovery from synthetic data

We first evaluate C2FAR-RNN's ability to recover the distribution of synthetic data, following [24, 28]. We generate data exactly as in [24], creating 500 series with a year's worth of hourly observations, each value sampled independently from a 3-component Gaussian mixture with weights $[0.3, 0.4, 0.3]$, means $[-3, 0, 3]$ and standard deviation of 0.4. To illustrate C2FAR-RNN's ability to model discrete data, we create an equivalent dataset where every element is drawn from a discrete uniform, $\mathcal{U}\{1, 10\}$.

On each dataset, we trained a C2FAR-RNN$_3$ model with 20 bins per level and hyperparameters based on *elec* experiments (below).

**Results.** In prior work, SQF-RNN [24] and IQN-RNN [28] struggled to recover the Gaussian mixture, but C2FAR can fit both distributions perfectly (Fig. 5). C2FAR is evidently the state-of-the-art in capturing complex, multi-modal densities without any prior information.

## 5.2 Empirical study on real-world data

**Datasets.** We evaluate C2FAR forecasting models on the following datasets:

- *elec*: hourly electricity usage of 321 customers [17], using *discretized* version from [35].
- *traff*: real-valued hourly occupancy from 0 to 1 for 862 car lanes [17], using version from [76].
- *wiki*, daily integer number of hits for 9535 pages, first used in [24], using same version as in [28].
- *azure*, (discrete) hourly usage of virtual machine (VM) flavors and flavor groupings (by tenant, deployment type, etc., in units of VCPUs or GB of memory) in a large public cloud, based on data from [15]. We release this dataset publicly as a paper supplement (see supplementary §C.1 for further details).

For *azure*, we use 20 days as training, 3 days for validation, and 3 final days for testing. Other splits are as in [54]. We forecast 24 hours for hourly series, 30 days for *wiki*, and use rolling evals as in [28].

**Metrics.** To evaluate point forecasts, we output the median of our forecast distribution and compute *normalized deviation* (ND) from true values. We evaluate probabilistic forecasts using *weighted quantile loss* (wQL), evaluated at forecast quantiles $\{0.1, 0.2 \ldots 0.9\}$, as in [49]. For non-probabilistic baselines Naïve and Seasonal-naïve (described below), note wQL reduces to ND, similarly to how CRPS reduces to absolute error [27, §4.2]. We also test the *calibration* and *sharpness* [71] of each system, measuring coverage of true values within particular percentiles (e.g. from 10% to 90%, coverage closer to 80% is better) and normalized width of this interval (lower is better). For a target coverage of X%, we refer to these metrics together as *CovX* (e.g. Cov80). Scores at extreme percentiles (e.g., Cov99%) help evaluate modeling of distribution tails.

**Baselines.** We compare our model to the following baselines:

- Naïve [32]: outputs the last-observed historical value at all forecast horizons
- Seasonal-naïve [32]: at each horizon, outputs the value at the most-recently-observed *season* matching the season at that horizon. We thus output the value at the most-recently-observed matching hour-of-day for hourly series, and matching day-of-week for the daily dataset (*wiki*).

Table 1: ND, wQL, Cov80, and Cov99 for our implementations (top), results from [49] (middle, denoted †) and [28] (bottom, denoted ‡), where available. In all cases, flat binned C2FAR-RNN$_1$ improves on DeepAR-Gaussian, while deeper C2FAR-RNN$_2$ likewise improves over C2FAR-RNN$_1$. Results are generally superior to prior state-of-the-art output distributions in RNN-based forecasting.

| | ND% | | | | wQL% | | | | Cov80% | Cov99% |
| | elec | traff | wiki | azure | elec | traff | wiki | azure | azure | azure |
|---|---|---|---|---|---|---|---|---|---|---|
| Naïve | 40.8 | 73.6 | 35.7 | 3.49 | 40.8 | 73.6 | 35.7 | 3.49 | - | - |
| Seasonal-naïve | 6.97 | 25.1 | 33.2 | 3.67 | 6.97 | 25.1 | 33.2 | 3.67 | - | - |
| ETS | 8.61 | 33.3 | 34.3 | 3.46 | 8.40 | 31.5 | 32.5 | 2.97 | 85.5/10.5 | 96.3/**20.3** |
| DeepAR-Gaussian | 7.05 | 16.1 | 43.8 | 3.60 | 5.60 | 13.7 | 54.7 | 3.06 | 89.9/16.9 | 98.0/37.7 |
| C2FAR-RNN$_1$ | 6.14 | 13.0 | 24.6 | 2.95 | 4.87 | 10.7 | 21.3 | 2.41 | 83.6/**8.3** | 98.5/32.2 |
| C2FAR-RNN$_2$ | 6.09 | **12.9** | 24.2 | 2.86 | 4.83 | **10.6** | **21.0** | 2.31 | **79.0**/8.5 | 98.4/29.1 |
| C2FAR-RNN$_3$ | **6.00** | 13.3 | **24.1** | **2.77** | **4.76** | 10.9 | **21.0** | **2.27** | 86.0/8.9 | **98.6**/32.7 |
| DeepAR-Binned† | 8.21 | 23.2 | 94.6 | - | 6.47 | 18.8 | 84.7 | - | - | - |
| DeepAR-StudentT† | 6.95 | 14.6 | 26.9 | - | 5.71 | 12.2 | 23.8 | - | - | - |
| IQN-RNN‡ | 7.40 | 16.8 | **24.1** | - | - | - | - | - | - | - |
| SQF-RNN‡ | 9.70 | 18.6 | 32.8 | - | - | - | - | - | - | - |
| DeepAR-StudentT‡ | 7.80 | 21.6 | 27.0 | - | - | - | - | - | - | - |

- ETS [31]: probabilistic state space models based on exponential smoothing, as implemented in `statsmodels` [56]. For each dataset, we tune (on validation data) whether to include *seasonality* and *trend* (*damped* or *undamped*) terms, and whether to use *estimated* or *heuristic* initialization.

- DeepAR-Gaussian: our implementation of DeepAR with a Gaussian output distribution; Gaussian outputs are common in practice, even on discrete data [37, 55, 14]

**Tuning.** Deeper C2FAR models include shallower models as special cases. We therefore address the question: does C2FAR's improved modeling outweigh the wider tuning required? For fair comparison, we establish a *parameter budget* [40] of 1M parameters; after sampling other hyperparameters, we restrict the possible number of LSTM hidden units so that this cap is enforced. We also restrict the number of tuning trials (on validation data) to 100 for each system.[4] We tune directly for normalized deviation via Optuna [1], with TPESampler [8], using MedianPruning and early stopping.

**Results**. Table 1 confirms that deeper C2FAR models improve over flat binnings. In only one case (*traff*) did a C2FAR model not improve over flat binning. In this case, C2FAR-RNN$_3$, with a larger tuning search space, also failed to find a superior setting of hyperparameters on validation data; going forward, using the same number of bins at each level (i.e., a constant $K$) could enable deeper C2FAR models without additional tuning. Contrary to prior work, we also find flat binning much more effective than standard parametric distributions. We attribute this difference to our use of systematic tuning; prior work used a fixed 1024 bins [49], while our tuner often selected quite fewer bins for C2FAR-RNN$_1$ (supplemental Table 6). In terms of both standard and extreme percentiles (Cov80% and Cov99% in Table 1), we find C2FAR models are both better calibrated than DeepAR-Gaussian (being twice as close to the desired coverage) while also having a sharper prediction interval width.

Seasonal-naïve is surprisingly competitive with DeepAR-Gaussian (except on *traff*, which has both daily and weekly seasonality). This has been observed previously (e.g., Table 1 in [2]); indeed, one of the authors of DeepAR and GluonTS has remarked that "a seasonal naive model performs almost as well as DeepAR or other deep models on [*elec* and *traff*], and without additional covariates, I suspect it's almost impossible to perform significantly better in terms of point predictions" [23]. But our results show that C2FAR *can* significantly improve over DeepAR and Seasonal-naïve without using additional covariates, but rather with a better output model.

Analyzing the distributions, we find C2FAR models are able to achieve good *precision* (having many intervals), while simultaneously learning smooth distributions (Fig. 6). Flat binnings must first *learn* "that a value of 128 is close to a value of 127 or 129" [53], while for C2FAR models, intervals will implicitly be close in probability because they are in the same coarser bins.

---

[4]In the prior work that mentions hyperparameter tuning, it is common to compare systems tuned with a fixed number of tuning trials, whether as part of grid search [77, 78, 51, 54] or other procedure (e.g., hyperopt in [12], random search in [38]). Training times are on the same order for all our trained systems (supplemental Table 7).

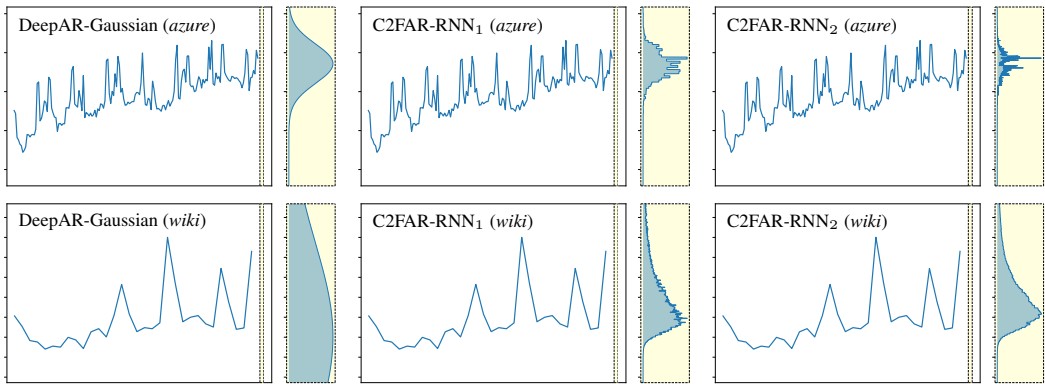

Figure 6: Distributions over possible next values. To cover likely values, Gaussians (left) also cover many unlikely ones. Flat binnings (middle) are optimized to either use fewer bins, but suffer in precision (top), or many bins but suffer in noise (bottom). C2FAR-RNN$_2$ (right) is able to place high probability on a repeat of the previous value (top) or generate smooth distributions (bottom).

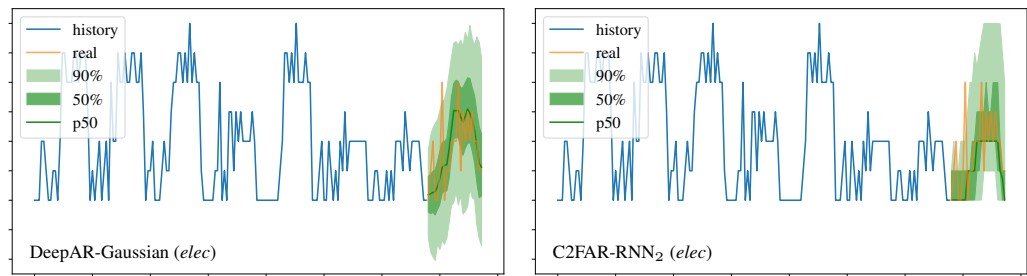

Figure 7: Forecast for an *elec* time series. DeepAR-Gaussian (left) yields a reasonable p50, but its lower percentiles span too low. C2FAR-RNN$_2$ yields better lower quantiles. C2FAR percentiles also suggest high-fidelity roll-outs, i.e., samples that closely mimic the discretized dynamics of the series.

Autoregressive models have a known disconnect between training, when true values are used as autoregressive inputs, and generation, when samples are used [5]. Our results suggest that such error accumulation may be especially problematic when a model is trained on discrete or mixed data, but uses standard continuous parametric outputs (as in DeepAR-Gaussian). Precise C2FAR models not only provide better forecast quantiles, they enable higher-fidelity samples to be recursively fed back in as autoregressive inputs (Fig. 7). Analyzing error by forecast horizon (supplemental §C.8), we find that at shorter horizons, DeepAR-Gaussian often performs similarly to C2FAR, but at later horizons, the gap widens. So while other solutions to error accumulation exist [65, 71, 73], one effective approach is evidently to generate and recurse on higher-fidelity outputs using C2FAR.

## 6   Conclusion

We presented C2FAR, a new method for density modeling. C2FAR reduces the generative process to a sequence of classifications over a hierarchical, discretized representation, with special handling of data outside the binning range. C2FAR can be applied to a variety of neural architectures, including RNN-based probabilistic forecasting, where it achieves state-of-the-art results when recovering synthetic distributions and forecasting real-world data. We show binned models (whether flat or, especially, coarse-to-fine) are superior to standard distributions — if binning precision is tuned.

Analysis shows that C2FAR can successfully model complex, multi-modal densities, in real or discrete data, without any prior information. This flexibility enables improved modeling for a variety of time series use cases, including forecasting, anomaly detection, interpolation, compression, denoising, and generating high-fidelity samples. It also enables development of large-scale forecasting models trained on diverse datasets, i.e., it is a step toward a universal neural forecaster.

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
