# OpenReview forum: "C2FAR: Coarse-to-Fine Autoregressive Networks for Precise Probabilistic Forecasting"
_NeurIPS.cc/2022/Conference — NeurIPS 2022 Accept_

### Official Review · Reviewer_sLp7 · 2022-07-10

**Rating:** 6
**Confidence:** 5
**Soundness:** 3 good
**Presentation:** 2 fair
**Contribution:** 2 fair

**Summary:**

The authors propose a new parametrize of the predictive distribution of probabilistic forecasting models. The build on prior work proposing binning or mixtures of uniform distributions (with parametric tails). They propose to avoid the pitfall of flat binning, where increasing the precision of the distribution requires to drastically increase the number of bins and so the number of output parameters of the neural network. They compare their method to existing work and show improvements on standard datasets.



**Questions:**

- How does one obtain a wQL% with the naive baseline, as the naive baseline does not provide a predictive distribution?
- You mention using your implementation of DeepAR. Why not use the open source implementation of DeepAR in GluonTS?

**Limitations:**

The author discuss exhaustively the limitations of their work, which is to be called out. They discuss the complexity and the efficiency of the method, I can see indeed that the efficiency depends a lot on the implementation and I would have liked to get to see their implementation.

**Strengths And Weaknesses:**

Strengths:
- The core idea of the paper is very good. It is a simple and efficient mechanism to solve the niche but real problem of the increase in the bins requiring an increase in the number of output parameters. Having a finer grid of bins through auto-regressive additions, is very good idea.
- I want to call out the honesty of the authors in the presentation of their work, their are openly vocal about the limitations of the work, which is something that should be praised.
- The author make a very valid point on the discretization of time series in general in the section 2.2.


Weaknesses:
A few things are not explained or laid out very clearly in the current version of the paper, and I think that the paper would benefit a lot from having these written out more clearly:
- I am not very clear on the parametrization of the neural network for p(z_i|x,z_{<i}). What kind of neural network is being used?
- How is the NLL computed at training time? (I wonder if one could optimize the code since only one value is required, that of the bin where the data point falls) I think that a pseudo code would be helpful to understand how this is being done.
- In the evaluation, please make clear what C2FAR-RNN1 is, and what the others are. I only saw in the appendix that if corresponds to K = 1.

- I would like to point out that I could verify all these points by looking at the code, but the authors do not provide the code. I see this as a major issue and would want the code to be open sourced if the paper is to be accepted.

I have difficulties understanding how the Pareto tails work exactly. In general, and compared to [17].
- alpha_1 and alpha_2 are outputs of the last layer, are they values on the observed space or quantiles?
- If they are values on the observed space, how is the density corresponding to the Pareto tail estimated?
- Or, does the maximum bin value move with alpha_1? If so, how does the network cope with the changing bin positions?
- Does having these tails force to compute the full densities?
- Are any of the experiments done with the tails? Have you evaluated the calibration of the tails?

Weaknesses in the evaluation:
- I would like to point out that DeepAR-Binned is not uniform bins but simply discretized values. And with that, the paper does not compare to the main comparison partner [17] which has uniform bins (like the flat tiling of C2FAR) and Pareto tails.
- It would be interesting to see the evaluation on a wider range of probabilistic forecasting datasets, and potentially regression datasets.
- The is a natural trade-off between the number of bins K in each level and the number of levels B. It would be interesting to lay it out clearly and to evaluate what the optimal trade-off is.
- It would be interesting to see the comparison to more flexible distributions like normalizing flows?
- It would be good to show the results with standard deviation.
- Overall, given the small methodological contribution of the paper it would have been valuable to provide a more exhaustive evaluation.



[17] Elena Ehrlich, Laurent Callot, and François-Xavier Aubet. 2021. Spliced Binned-Pareto Distribution for Robust Modeling of Heavy-tailed Time Series

---

> ### Author Response · Authors · 2022-08-02
> **Response to Reviewer sLp7 (1/2)**
>
> Thank you so much for your detailed and valuable feedback, and your
> generous support for the paper's core idea.
>
> You have identified several things that could be explained more
> clearly, and provided the feedback that we need in
> order to address your concerns and improve the paper's overall
> presentation.
>
> ### Regarding lack of clarity in the evaluation
>
> > In the evaluation, please make clear what C2FAR-RNN1 is
>
> Our apologies, we discussed this too briefly in L236.  We will make this clearer by:
>
> - Having a bulleted list defining C2FAR-RNN1, C2FAR-RNN2, C2FAR-RNN3,
>   which will make it easier to refer to this information.
>
> - Explicitly reminding the reader here of the connection between
>   C2FAR-RNN-1 and Ehrlich et al. [17], as we discuss below.
>
> - Developing and using the K and B terminology throughout the paper, in particular starting with Fig. 1.
>
> > DeepAR-Binned is not uniform bins but
>   simply discretized values... the paper does not compare
>   to the main comparison partner [17] which has uniform bins (like the
>   flat tiling of C2FAR) and Pareto tails.
>
> We again apologize for this confusion regarding the meaning of
> C2FAR-RNN-1.  We do compare hierarchical C2FAR models to C2FAR-RNN-1,
> which, like [17], has (flat) uniform bins and Pareto tails.  Of course,
> C2FAR-RNN-1 goes beyond [17] in terms of tuning the binning precision,
> and is implemented with a RNN in order to facilitate
> comparison to prior work.  We will make the relationship to [17] clear in the paper.
>
> > Have you evaluated the calibration of the tails?
>
> Yes, we mention calibration on L259 and report coverage on
> "extreme quantiles [i.e., Cov99%]" (L279) in
> Table 1.  We should have explicitly noted in the paper that one
> objective of this evaluation was to assess the modeling of the tails.
>
> > How does one obtain a wQL% with the naive baseline?
>
> Good question.  For point predictions, wQL reduces to
> ND, similarly to how CRPS reduces to absolute error
> (see Sec. 4.2 of [Gneiting and Raftery,
> 2007](https://sites.stat.washington.edu/raftery/Research/PDF/Gneiting2007jasa.pdf)).
> We will add a brief note about this in the paper.
>
> Note you see this in other forecasting papers, without explanation, e.g. in [the
> GluonTS
> paper](https://www.jmlr.org/papers/volume21/19-820/19-820.pdf), Table
> 1, they report "mean quantile loss" for `seasonal-naive`.
>
> > Why not use the open source implementation of DeepAR in GluonTS?
>
> 1. Convenience: our production forecasting system uses PyTorch and we
> can therefore re-use our existing tuning pipelines, data preparation,
> evaluation modules, etc. for DeepAR.
>
> 2. Experimental rigour: by using the same code for DeepAR and
> C2FAR-RNN, we can ensure these systems differ only in the way described in
> Sec. 4.2 of the paper.
> The GluonTS implementation may have hidden impactful differences from our own setup,
> e.g. the way training windows are sampled, the way timestamp-related covariates are created,
> whether there are scheduled reductions in learning rates during training, etc.
>
> ### Regarding computation of NLL
>
> > How is the NLL computed at training time?
>
> In Footnote 1 we mention:
>
> *"full C2FAR densities are never explicitly manifested when training
>  or predicting; we need only compute the top-level distribution, then
>  finer-grained distributions within each observed (or sampled) coarser
>  bin..."*
>
> We will promote this footnote to be part of the main paper text, and
> clarify the point further by adding the following:
>
> ```
> Indeed, we may regard the C2FAR binning as a K-ary tree of height B.
> At each node in the tree, we determine which of the K bins we fall
> into at that level, and follow the chosen branch to the sub-tree at
> the next level.  We need only evaluate the softmax probabilities (and
> backpropagate gradients) for the B nodes on the path from the root to
> the leaf of the tree.  Since the height of the tree, B, is logarithmic
> (in base K) over the total number of intervals, we compute
> exponentially fewer outputs with C2FAR compared to flat binnings.  The
> final likelihood additionally requires computing the probability at
> the B+1 level according to the corresponding parametric distribution
> (Uniform or Pareto) as given in Eqn (2).
> ```
>
> > I think that a pseudo code would be helpful
>
> Excellent idea.  We will provide some simple (non-vectorized) pseudocode to illustrate this mechanism.
>
> > Does having these [Pareto] tails force to compute the full densities?
>
> No.  Hopefully this will be clear after we add the clarifying paragraph above.
>
> ### Regarding availability of the source code
>
> > I could verify all these points by looking at the code
>
> We are working to share an open-source implementation of the code.  Our
> organization recognizes that open sourcing could
> increase the impact of the work, and code has been released with our previous
> publications.  However unfortunately the bureaucratic process begins
> only after paper acceptance.

---

> > ### Author Response · Authors · 2022-08-02
> > **Response to Reviewer sLp7 (2/2)**
> >
> > ### Regarding the optimal trade-off between K and B
> >
> > > There is a natural trade-off between the number of bins K in each
> >   level and the number of levels B. It would be interesting to ... evaluate what the optimal trade-off is.
> >
> > Consider a $B$-level C2FAR binning with $K$ bins at each level, with $I=K^B$ total intervals.  In C2FAR, we need to compute $K * B$ outputs ($B$ softmaxes with $K$ values each), i.e., $K*log_K(I)$.  From a discretization perspective, the optimal $K$ would actually be 2, as then, rather than a softmax, we could use a single logistic function output that is thresholded at 0.5 to select the high or low bin, and thus only compute $log_2(I)$ outputs.
> >
> > However, the ultimate complexity of C2FAR also depends on the neural architecture. For C2FAR-RNN, we operate $B$ separate LSTMs, and so it is typically more efficient to use a slightly higher value of $K$ (but not so high that computing output probabilities dominates the complexity) and consequently a lower value of $B$. (Please also see our note on algorithmic complexity in response to Reviewer coLS)
> >
> > We will definitely add some notes regarding this to the paper.  Thanks a lot for this suggestion!
> >
> > ### Regarding the parameterization of the neural network for p(z_i|x,z_{<i})
> >
> > > I am not very clear on the parametrization of the neural network for
> >   p(z_i|x,z_{<i}). What kind of neural network is being used?
> >
> > Yes, we need to make this clearer: the relevant line from the paper is
> > L129 ("we parameterize these distributions using neural networks with a
> > softmax output layer...").  In fact, at this point we are speaking generically, and only at L133
> > do we discuss some of the specific neural architectures that can be
> > used here.  We will provide a forward pointer on L129 to the paragraph L133-L138.  In that paragraph, and
> > in Fig. 2, we will clarify that here we are envisioning a collection
> > of $B$ *feedforward neural networks* for the task of (univariate)
> > regression, in contrast to the $B$ recurrent neural networks adopted
> > in Sec. 4.2 and Fig. 4 for (sequential) forecasting.
> >
> > ### Regarding how the Pareto tails work
> >
> > > I have difficulties understanding how the Pareto tails work
> >   exactly.
> >
> > We wil clarify that "weighted" Pareto tails are not
> > "spliced" into the distribution at fixed user-defined quantiles, as they are
> > in [17].  In C2FAR, Pareto tails provide the probability density
> > function of values in the extreme bin, *given the values are in the
> > extreme bin*.  Each Pareto density integrates to 1,
> > but is dynamically "weighted" by the probability of being in the corresponding extreme bin.
> > Since the position of all bins are fixed, the $a$ and
> > $b$ values for the Paretos in Eqn. (2) are always known a priori.
> > Thus, when training or predicting, the only parameters that are dynamic are the
> > shape parameters ($\alpha_1$ and $\alpha_2$).
> >
> > > alpha_1 and alpha_2 ... are they values on the observed space or quantiles?
> >
> > We will clarify in the paper that we use a Type I Pareto Distribution, where
> > $\alpha_i$ is the shape parameter of the distribution.  Our network
> > outputs the shape parameter in a manner analogous to how DeepAR
> > outputs the mean and variance parameters of a Gaussian distribution.
> > The tail distribution is defined in observed space, but note that once we move to forecasting (Sec. 4),
> > the observed values will be normalized using the conditioning range.
> >
> > > how is the density corresponding to the Pareto tail estimated?
> >
> > The Pareto's density is noted in Eqn. (2): it is determined by both
> > $\alpha_i$ (as discussed above) and the scale parameter of the
> > distribution, $a$ or $b$, which is the fixed starting point of the extreme
> > bin (as also mentioned above).
> >
> > > does the maximum bin value move with alpha_1?
> >
> > No, the bin locations do not move.
> >
> > ### Regarding additional experiments
> >
> > > it would have been valuable to provide a more exhaustive evaluation.
> >
> > Your point is well taken, however we do believe that our
> > evaluation was fairly extensive, being both
> > wide (4 real-world and 2 synthetic datasets) and deep (improving the
> > rigour of forecasting evaluations).  For example, we use a parameter
> > budget to ensure fair comparison of systems; such a concept has not
> > been used in prior forecasting work.  Moreover, we carefully describe
> > and use a systematic hyperparameter tuning method, tuning both our own
> > proposed system and the DeepAR and flat-binning baselines.  Note that the
> > prior work on binning does not use such a systematic
> > approach, e.g. in Rabanser et al. 2020 [39], "principled
> > hyperparameter tuning" was left as future work.  In this way, our
> > evaluation is a strength of the paper (as noted by the other
> > reviewers), and is valuable to the forecasting community as it helps
> > establish the benefits of binning in general.

---

> > > ### Comment · Reviewer_sLp7 · 2022-08-10
> > > **Response to authors**
> > >
> > > I thank the authors for their detailed response, and for their response to the other authors.
> > > They were able to provide a lot of clarifications to the method in their answer, and I hope that they will be able to improve their revision accordingly.
> > > With this I raise my score from 5 to 6.

---

### Official Review · Reviewer_coLS · 2022-07-10

**Rating:** 7
**Confidence:** 4
**Soundness:** 3 good
**Presentation:** 3 good
**Contribution:** 3 good

**Summary:**

The paper proposes C2FAR, a new method for probabilistic univariate forecasting. C2FAR sequentially discretizes the time-series in bins, from coarse to fine resolution, using an RNN to learn temporal relations similar to the DeepAR model. At each layer, a shared RNN model further divides each bin into finer intervals based on covariates and values of different layers. Forecasts are produced by drawing samples conditional on the learned parameters. The paper compares the proposed method against other RNN probabilistic architectures, such as DeepAR, IQN-RNN, and SQF-RNN, on several benchmark datasets.

**Questions:**

- Inference (prediction) times are not reported or mentioned in the paper. Can you provide average inference times for different datasets?
- Have you evaluated C2FAR on longer horizons? Since forecasts are produced recursively, errors will accumulate, degrading the performance considerably.

**Limitations:**

Limitations are discussed, and I do not identify additional potential negative social impacts.


**Strengths And Weaknesses:**

Strengths:
- Novel method for probabilistic univariate forecasting, based on sequential hierarchical binning.
- SoTA performance on 4 benchmark datasets, compared to other RNN-based models.
- The shared model between bins of each layer allows for exponentially expressiveness with linear cost.
- The last layer can handle extreme values thanks to the Pareto-distributed tails.
- Comprehensive ablation and stability experiments. The experimental setting is very clear and thoroughly explained.

Weaknesses:
- Only RNN architectures are used as baselines. While RNN is widely used, other architectures have shown superior performance in multiple forecasting tasks, especially for multi-step forecasting. Authors should include other models in the experiments, including simple baselines such as seasonal-naive, ETS, or ARIMA.
- C2FAR requires more training time and resources than baselines. Training time should be also compared against other baselines.

---

> ### Author Response · Authors · 2022-08-02
> **Response to Reviewer coLS (1/2)**
>
> Thank you very much for your thoughtful and constructive feedback, and your kind words regarding
> our novel method and clear and comprehensive evaluation.
>
> ### Regarding evaluation of other forecasting models
>
> > While RNN is widely used, other architectures have shown superior
> performance in multiple forecasting tasks, especially for multi-step
> forecasting.
>
> Yes, this is a fair point.  Indeed, RNNs still dominate industrial
> approaches to forecasting, including [Amazon
> Forecast](https://docs.aws.amazon.com/whitepapers/latest/time-series-forecasting-principles-with-amazon-forecast/time-series-forecasting-principles-with-amazon-forecast.pdf),
> and our own production forecast service.  Moreover, even 1-step-ahead
> forecasting still has important applications, although we focus on 24-step-ahead and
> 30-step-ahead forecasts in this paper.  That being said, C2FAR is
> compatible with other sequence architectures and we are especially
> interested in evaluating transformer-based systems with and without
> the C2F decoding.  We will certainly pursue these further experiments
> in advance of the camera paper deadline.
>
> > Authors should include other models in the experiments, including
>   simple baselines such as seasonal-naive, ETS, or ARIMA.
>
> This is also a very good suggestion.  We did include results for naive
> in the paper (Table 1).  However, based on your suggestion, we are now evaluating
> seasonal-naive, ETS, and ARIMA as well (systematically tuning them as
> needed as we did our other baselines).
> Note that ETS scores were previously reported on `electricity`,
> `traffic`, and `wikipedia` in Gouttes et al. 2021, but were
> significantly worse than those of IQF-RNN.
>
> In fact, we have already completed our evaluation of seasonal-naive, and it is surprisingly competitive
> with DeepAR-Gaussian on `elec` and `azure` - although not as good as C2FAR!
>
> **Table 1 (showing ND% and wQL% results only)**
> |                    | ND%    | ND%     | ND%    | ND%     | wQL%   | wQL%    | wQL%   | wQL%    |
> |--------------------|--------|---------|--------|---------|--------|---------|--------|---------|
> |                    | _elec_ | _traff_ | _wiki_ | _azure_ | _elec_ | _traff_ | _wiki_ | _azure_ |
> | naive              | 40.8   | 73.6    | 35.7   | 3.49    | 40.8   | 73.6    | 35.7   | 3.49    |
> | **seasonal-naive** | 6.97   | 25.1    | -      | 3.67    | 6.97   | 25.1    | -      | 3.67    |
> | DeepAR-Gaussian    | 7.05   | 16.1    | 43.8   | 3.60    | 5.60   | 13.7    | 54.7   | 3.06    |
> | C2FAR-RNN1         | 6.14   | 13.0    | 24.6   | 2.95    | 4.87   | 10.7    | 21.3   | 2.41    |
> | C2FAR-RNN2         | 6.09   | 12.9    | 24.2   | 2.86    | 4.83   | 10.6    | 21.0   | 2.31    |
> | C2FAR-RNN3         | 6.00   | 13.3    | 24.1   | 2.77    | 4.76   | 10.9    | 21.0   | 2.27    |
> | DeepAR-Binned(y)   | 8.21   | 23.2    | 94.6   | -       | 6.47   | 18.8    | 84.7   | -       |
> | DeepAR-StudentT(y) | 6.95   | 14.6    | 26.9   | -       | 5.71   | 12.2    | 23.8   | -       |
> | IQN-RNN(z)         | 7.40   | 16.8    | 24.1   | -       | -      | -       | -      | -       |
> | SQF-RNN(z)         | 9.70   | 18.6    | 32.8   | -       | -      | -       | -      | -       |
> | DeepAR-StudentT(z) | 7.80   | 21.6    | 27.0   | -       | -      | -       | -      | -       |
>
> Note seasonal-naive has previously been found to be competitive with neural models (e.g. Table 1 in [Alexandrov et al., JMLR, 2020](https://www.jmlr.org/papers/volume21/19-820/19-820.pdf)).  One of the authors of the DeepAR and GluonTS papers [has previously said](https://github.com/awslabs/gluon-ts/discussions/1180?sort=new#discussioncomment-169433):
>
> *"A seasonal naive model performs almost as well as DeepAR or other deep models on these data sets [`elec` and `traffic`], and without additional covariates, I suspect it's almost impossible to perform significantly better in terms of point predictions."*
>
> But the above results show that C2FAR *can* significantly improve over DeepAR and seasonal-naive without using additional covariates, but rather with a better output model.  We should definitely add discussion along these lines in the paper. Thanks a lot!
>
> > Training time should be also compared against other baselines.
>
> Yes, we will report training times of the baselines.  We provide
> training times for all evaluated systems in supplemental Table 7, but
> we absolutely should have included this in the main paper.  We may also
> include `naive` and `seasonal-naive` in this table and note training time is `N/A` in order to
> emphasize that simple baselines do not have training overhead.

---

> > ### Author Response · Authors · 2022-08-02
> > **Response to Reviewer coLS (2/2)**
> >
> >
> > ### Regarding inference times
> >
> > > Inference (prediction) times are not reported or mentioned in the
> >   paper. Can you provide average inference times for different
> >   datasets?
> >
> > Yes, we will do this.  Right now we do note in L174 of the supplement:
> > "testing time roughly follows a similar pattern [to Table 7 for
> > training times]", but we will include the following complete table in
> > the camera-ready paper.
> >
> > **Table: Time per 100 forecasts in *seconds* (running on NVIDIA Tesla P100) by top system on test set. All systems ran with common test batch sizes (60 for daily wiki, 22 for hourly datasets) and number of samples (500) for each dataset.**
> >
> > |                 | Elec | Traffic | Wiki | Azure | Average |
> > |-----------------|------|---------|------|-------|---------|
> > | DeepAR-Gaussian | 1.28 | 1.57    | 0.55 | 2.72  | 1.53    |
> > | C2FAR-RNN1      | 3.73 | 3.79    | 1.71 | 1.34  | 2.64    |
> > | C2FAR-RNN2      | 4.22 | 4.12    | 1.50 | 1.68  | 2.88    |
> > | C2FAR-RNN3      | 4.81 | 4.45    | 1.92 | 4.73  | 3.98    |
> >
> > Note we also plan to add a discussion of the algorithmic complexity of these models, in Section 4.  Let $H$ be the number of hidden units in the RNN.  The complexity of C2FAR-RNN1 is dominated by $K * H$, where $K$ is the total number of bins in the flat binning.  The complexity of DeepAR-Gaussian is dominated by $H * H$, while B-level C2FAR models are dominated by $B * H * H$, i.e., the cost of running $B$ RNNs in parallel.  The times in the above table are well explained by these observations, given the corresponding values for $H$, $K$, and $B$ provided in supplemental Table 6.
> >
> > ### Regarding longer horizons
> >
> > > Have you evaluated C2FAR on longer horizons? Since forecasts are
> >   produced recursively, errors will accumulate, degrading the
> >   performance considerably.
> >
> > In terms of number of steps, we have not evaluated C2FAR beyond the
> > 30-days-ahead predictions for Wiki.  Internally, we perform
> > longer-*duration* forecasts with C2FAR, e.g., 13-week forecasts, but
> > at the weekly granularity (so only 13-steps-ahead).  We like the idea
> > of testing C2FAR when forecasting a very high number of steps.  We actually hypothesize
> > that C2FAR will work better than other auto-regressive RNNs because
> > the samples that are being fed back in as inputs are more realistic
> > (we alluded to this when discussing "high-fidelity samples" in L286-L289).  We
> > are testing this now, both evaluating the accuracy at each horizon for
> > the existing datasets and models, and trying new datasets with longer
> > prediction ranges.  Good suggestion.

---

> > > ### Comment · Reviewer_coLS · 2022-08-08
> > > **Additional results**
> > >
> > > Thank you for adding additional baselines and times, these improve the empirical results considerably. I will rise the score to accept.

---

### Official Review · Reviewer_BP7i · 2022-07-17

**Rating:** 7
**Confidence:** 4
**Soundness:** 3 good
**Presentation:** 3 good
**Contribution:** 3 good

**Summary:**

This paper introduces a mechanism, C2FAR, to parametrize a conditional distribution via successive hierarchical binnings parametrized by an autoregressive function (here, a neural network). This parametrization can reportedly maintain expressibility in high dimensional binned distributions with logarithmically many parameters required. The authors present a thorough empirical study of their method in the backdrop of time series forecasting, where C2FAR-RNN (their hierarchical--coarse-to-fine--binned distribution on top of DeepAR) performs favorably both in forecast accuracy and calibration.

**Questions:**

- The authors allude to limited precision in most forecasting data. I am not sure if this is really a justification for the method? Note that this is different than significant "atoms" of probability observed on 0% or 100%--reminiscent of zero-inflated distributions which are more similar to the authors' construction.
- In Eq (1), should the lhs be $p(\mathbf{z} | \mathbf{x})$ instead?
- On L129: "Inputs may be represented with 1-hot-encodings or embedding layers" Which inputs are these? $\mathbf{x}$? If not, what is the difference implied between one-hot encoding and an embedding layer?
- L192.  During training, and when evaluating NLL of test sequences, all values are known and all distributions can be computed in parallel. I am not sure of this, as the backprop would have to be sequential in any case with an RNN?
- Why restrict the total number of hyperparameter optimization runs instead of total time? Are the training times on the same order? If so, please mention. Seeing as the sequential evaluation is the bottleneck in GPU implementations, then it may be that computation really takes B-times more time than DeepAR in training, which is unfair. I would urge the authors to report wall-clock training times of their method against DeepAR in the same number of bins.


**Limitations:**

Other than the last question mentioned above, I do not believe there are significant limitations of the paper.

**Strengths And Weaknesses:**

The paper is well-written, well-organized and gives a good review of comparable methods in forecasting. To the best of my knowledge, the introduced techniques are novel, although I couldn't help comparing them to hierarchical softmax/categorical distributions used in NLP and wondered if any analogies could be drawn from there. In the forecasting domain, I believe the results posted by the authors are significant and advance the state of the art. Despite the simplicity of the ideas proposed, I believe the paper is insightfully written and benefits from a well-thought-out experimental set-up. Moreover, I believe C2F can have impact in other related areas of probabilistic deep learning.

---

> ### Author Response · Authors · 2022-08-02
> **Response to Reviewer BP7i (1/2)**
>
> Thank you very much for your insightful and helpful review, and your
> kind words regarding our key idea, clarity of presentation, and
> thorough evaluation.
>
> ### Regarding the connection to the hierarchical softmax used in NLP
>
> >the introduced techniques are novel, although I couldn't help
> comparing them to hierarchical softmax/categorical distributions used
> in NLP and wondered if any analogies could be drawn from there
>
> We really like this suggestion and will definitely expand the related
> work to discuss this.  For example, we may add the following:
>
> ```
> A hierarchical decomposition of a categorical distribution has
> been used previously for large-vocabulary language modeling [Morin &
> Bengio 2005, Mnih & Hinton 2008].  As with C2FAR, the key benefit is
> that exponentially fewer computations are required than when
> representing the full vocabulary with a flat softmax.  Later work
> pursued further efficiencies via variable-length representations
> (based on Huffman coding) [Mikolov et al., 2013]; whether such
> enhancements could prove effective when modeling continuous and
> ordinal data with C2FAR merits further investigation.
> ```
>
> ### Regarding fixing the number of optimization trials for each system
>
> > Why restrict the total number of hyperparameter optimization
> runs instead of total time? Are the training times on the same order?
>
> First of all: yes, training times are on the same order.  We provide
> training times for all evaluated systems in supplemental Table 7, but
> we absolutely should have included this in the main paper.
>
> We also should have it made clear that (in those cases where
> hyperparameter tuning is explicitly described) it is standard for
> forecasting evaluation to use a fixed number of tuning trials, whether
> as part of grid search (e.g. [Informer](https://arxiv.org/abs/2012.07436), [MixSeq](https://proceedings.neurips.cc/paper/2021/file/6b5754d737784b51ec5075c0dc437bf0-Paper.pdf), [End-to-End Learning of
> Coherent Forecasts](http://proceedings.mlr.press/v139/rangapuram21a/rangapuram21a.pdf), [High-Dimensional Forecasting with Low-Rank Copula Processes](https://proceedings.neurips.cc/paper/2019/file/0b105cf1504c4e241fcc6d519ea962fb-Paper.pdf), etc.) or other search procedure (hyperopt in [NHiTs](https://arxiv.org/abs/2201.12886), random
> search in [Temporal Fusion Transformers](https://arxiv.org/abs/1912.09363), etc.).
>
> To control for the potential unfairness of more complex models, our
> paper introduces the concept of the "parameter budget" to the
> forecasting community.  Adopting a time budget is also a good idea,
> although we may need to deploy more sophisticated optimization
> algorithms (e.g. bandit-based algorithms like
> [Hyperband](https://arxiv.org/abs/1603.06560)), so that a trial with
> a randomly-sampled especially-slow learning rate can be paused or pruned so
> that it does not use up disproportionately large portions of the time budget.
>
> It may also be worth mentioning how C2FAR may also naturally be
> tuned in light of budgets on *inference* (whether time or memory).
>
> > I would urge the authors to report wall-clock training times of
> their method against DeepAR in the same number of bins.
>
> Indeed, for the systems in supplemental Table 7, sometimes the optimal
> C2FAR system uses quite more, sometimes quite fewer total intervals
> than the flat DeepAR-Binned method (as shown in supplemental Table 6).
> Following your suggestion, we plan to repeat the training/inference
> timing while fixing either the number of bins in each level, or the
> total number of intervals.  Good suggestion.
>
> ### Regarding parallelism
>
> > L192. During training, and when evaluating NLL of test
> sequences, all values are known and all distributions can be computed
> in parallel. I am not sure of this, as the backprop would have to be
> sequential in any case with an RNN?
>
> Yes, good catch.  It is only parallel in the sense that you do not
> need to evaluate the coarser softmaxes before evaluating the finer softmaxes,
> i.e., it can be parallelized within each time step.  As such, this
> point properly belongs back in Section 3 before we introduce the RNN.
>
> ### Regarding Eq (1)
>
> > In Eq (1), should the lhs be p(z|x) instead?
>
> Yes, right you are again.

---

> > ### Author Response · Authors · 2022-08-02
> > **Response to Reviewer BP7i (2/2)**
> >
> > ### Regarding limited precision
> >
> > > The authors allude to limited precision in most forecasting
> > data. I am not sure if this is really a justification for the method?
> > Note that this is different than significant "atoms" of probability
> > observed on 0% or 100%--reminiscent of zero-inflated distributions
> > which are more similar to the authors' construction.
> >
> > Yes, we were stepping a little too lightly here, and this section
> > needs a topic sentence to really clarify our (somewhat esoteric)
> > message:
> >
> > 1. that there is a need to model discretized data (for a
> > variety of reasons),
> > 2. simple discrete output distributions have proven inadequate, and
> > 3. there is a kind of "bug" (that prior work is often not aware of)
> > when applying powerful continuous models to discrete data directly
> > (I believe this is the "very valid point" mentioned by Reviewer sLp7
> > regarding this section).
> >
> > And yes it is a good point that 0% and 100% "atoms of probability"
> > should properly be described separately (as they occur beyond discrete
> > distributions), perhaps when we describe the Seeger et al. multi-stage
> > likelihood model ([45] in Sec. 2.3). We should mention approaches to
> > zero-inflation directly here as well (as they do in Seeger et al).
> > Thanks again.
> >
> > ### Regarding Embedding
> >
> > > On L129: "Inputs may be represented with 1-hot-encodings or
> > embedding layers" Which inputs are these? x? If not, what is the
> > difference implied between one-hot encoding and an embedding layer?
> >
> > Thanks for mentioning this.  We will clarify that here we are talking
> > about the autoregressive inputs z.  The point we are making (and which
> > we will also clarify) is that C2FAR itself is agnostic toward the
> > encoding of the inputs (whether one-hot encoded or embedded).

---

### Meta-Review · Area_Chair_3KJf · 2022-08-27

**Recommendation:** Accept
**Confidence:** Certain

**Metareview:**

The reviewers highlight the clarity of the writing, the effectiveness of the proposed approach resulting in SOTA performance, and in particular the comprehensive and clear empirical evaluation with ablation studies and stability experiments, as well as a laudable discussion of the limitations of the proposed approach. The authors were able to address reviewers' concerns around baselines and comparison partners during the discussion period.

**Award:**

No

---

### Decision · Program_Chairs · 2022-09-14

Accept